# What Is Currently the Role of TcPO2 in the Choice of the Amputation Level of Lower Limbs? A Comprehensive Review

**DOI:** 10.3390/jcm10071413

**Published:** 2021-04-01

**Authors:** Judith Catella, Anne Long, Lucia Mazzolai

**Affiliations:** 1Service de Médecine Interne et Vasculaire, Hopital Edouard Herriot, Hospices Civils de Lyon, 69003 Lyon, France; anne.long@chu-lyon.fr; 2Laboratoire d’Excellence du Globule Rouge (Labex GR-Ex), Sorbonne, 75015 Paris, France; 3Laboratoire Interuniversitaire de Biologie de la Motricité (LIBM) EA7424, Université Claude Bernard Lyon 1, Université de Lyon, 69008 Lyon, France; 4UMR 5305: Laboratoire de Biologie Tissulaire et Ingénierie Thérapeutique, CNRS/Université Claude Bernard Lyon 1, Institut de Biologie et Chimie des Protéines, 7 Passage du Vercors, 69367 Lyon, France; 5Service d’Angiologie, Département Coeur et Vaisseaux, Centre Hospitalier Universitaire Vaudois, 1011 Lausanne, Switzerland; lucia.mazzolai@chuv.ch

**Keywords:** TcPO2, chronic limb ischemia, amputation, peripheral arterial disease

## Abstract

Some patients still require major amputation for lower extremity peripheral arterial disease treatment. The purpose of pre-operative amputation level selection is to determine the most distal amputation site with the highest healing probability without re-amputation. Transcutaneous oximetry (TcPO2) can detect viable tissue with the highest probability of healing. Several factors affect the accuracy of TcPO2; nevertheless, surgeons rely on TcPO2 values to determine the optimal amputation level. Background about the development of TcPO2, methods of measurement, consequences of lower limb amputation level, and the place of TcPO2 in the choice of the amputation level are reviewed herein. Most of the retrospective studies indicated that calf TcPO2 values greater than 40 mmHg were associated with a high percentage of successful wound healing after below-knee-amputation, whereas values lower than 20 mmHg indicated an increased risk of unsuccessful healing. However, a consensus on the precise cut-off value of TcPO2 necessary to assure healing is missing. Ways of improvement for TcPO2 performance applied to the optimization of the amputation-level are reported herein. Further prospective data are needed to better approach a TcPO2 value that will promise an acceptable risk of re-amputation. Standardized TcPO2 measurement is crucial to ensure quality of data.

## 1. Introduction

Several strategies, such as distal percutaneous endovascular therapies, are available for the treatment of lower extremity peripheral arterial disease (LE-PAD); however, some patients still require major amputation [1].

Lower extremity amputation is generally reserved for situations where medical or revascularization options are no longer available, when significant tissue loss has occurred, or when optimal medical therapy and/or revascularization has failed. It is often considered as a last resort because it is associated with substantially higher mortality. In 2014, the rate of death after major amputation in the US was estimated to approximately 48% at 1 year and 71% at 3 years [2].

A minor amputation of lower limbs is defined as any procedure that results in an amputation up until the ankle level. A major amputation of lower limbs is defined as any procedure that results in an amputation one above the ankle joint [2]. It is further categorized based on its location relative to the knee: a below-knee-amputation (BKA) is an amputation affecting the tibia at any point below the knee and above the ankle; an above-knee-amputation (AKA) is an amputation above the knee affecting the femur at any level [2]. It is commonly admitted that the best prognosis of rehabilitation is encountered in case of preservation of the knee joint.


The purpose of pre-operative amputation level selection is to determinate the most distal amputation site with the highest healing probability without re-amputation. Then, the main concern is successful mobility.


Transcutaneous oximetry (TcPO2) is currently used to detect viable tissue with the highest probability of healing. TcPO2 is also used to define chronic limb ischemia and predict the ability of ulcers to heal [3]. Several factors affect the accuracy of TcPO2; nevertheless, surgeons rely on TcPO2 values to determine the optimal amputation level. At present, a consensus on the precise cut-off value of TcPO2 necessary to assure healing is missing. 


Background about the development of TcPO2, methods of measurement, consequences of lower limb amputation level, and the place of TcPO2 in the choice of the amputation level are reviewed herein. Finally, current recommendations and ways of improvement for TcPO2 performance applied to the optimization of the amputation level are reported.


## 2. Background

As early as 1793, John Abernathy observed that the human skin releases small amounts of carbon dioxide [4] and provided the first evidence that the skin is permeable to gases. In 1925, Heyrovsky demonstrated that by applying a proper voltage to a mercury electrode, the produced current was proportional to the electrolyzed substance tension in the surrounding liquid (i.e., O_2_ in TcPO2) [4]. Polarography has applications in numerous fields, and Jaroslav Heyrovský won the Nobel Prize in Chemistry in 1959 for his research.

Baumberger and Goddfriend demonstrated that the cutaneous oxygen tension values reflect arterial partial pressure of oxygen (PaO_2_) values [4]. By 1950, scientists were using platinum electrodes to measure the O_2_ tension of living tissues. In 1954, Leland Clark, Jr., developed a new polarographic electrode in which the platinum cathode was covered with a semipermeable membrane to produce a more uniform O_2_ layer [5].

In 1972, Huch et al. reported the use of heated Clark electrodes as a practical method to monitor skin surface (transcutaneous) pO_2_. These instruments were first applied clinically to monitor O_2_ in neonates during and after delivery [6]. TcPO2 was found to estimate PaO_2_ values closely, but when the blood flow was compromised, TcPO2 reflected only local O_2_ delivery [7].

Esato el al. showed a negative correlation between the grade of Fontaine classification and the TcPO2 with significant difference between grade II and IV on 38 lower limbs [8]. Moreover, decrease in TcPO2 in LE-PAD patients was well correlated with ankle systolic pressure [9]. TcPO2 values depends on LE-PAD Fontaine classification, site of vascular obstruction, and hemodynamic compensation by calf arteries patency. Franzeck et al. proposed using TcPO2 to determinate the optimal amputation level in patients with LE-PAD in 1982, after noting a decrease TcPO2 in lower extremities of these patients [10]. 

## 3. The TcPO2 Technique

TcPO2 measurement is a metabolic test while ankle brachial index, plethysmography, and Doppler systolic pressure are hemodynamic indexes. Wyss et al. showed that when PaO_2_ is normal or near normal, a curvilinear relationship exists between TcPO2 values and local perfusion pressures. Therefore, when the perfusion pressure is high, a small decrease in perfusion pressure causes only a small decrease in TcPO2. Conversely, when the perfusion pressure is low, a small decrease in perfusion pressure causes a much greater decrease in TcPO2 which may even reach 0 mmHg despite blood flow evidenced using other techniques [11]. Furthermore, it was pointed out that a null TcPO2 value did not indicate that no oxygen is reaching the tissue but rather that the oxygen delivery equals or exceeds the metabolic consumption of the skin [12]. 

It has been shown that the technique can be easily managed by hospital technicians, regular nursing, and/or junior medical staff. A site reading requires an average of 35 min. The costs are well within the capability of an average general hospital in developed countries, and the running costs are considered negligible [13]. Transcutaneous oxygen tension measurement is easily and noninvasively obtained and can be applied to all patients irrespective of Doppler signals, non-compressible vessels, or painful lesions [14].

The sensor is applied to the skin by a flat, double-sided adhesive ring. The oxygen diffuses according to its pressure gradient from the capillary loops through the avascular epidermis towards the skin surface electrode. In clinical practice, there is no consensus regarding the exact position of the electrode which is determined for each case by the operator considering the ischemic status of the lower limbs and local habits. 

Consistent and reproducible values are obtained when the required local arterialization is achieved by heating the oxygen electrode itself [15]. An electrode heated to 45 °C transmits a temperature of about 43 °C on the skin, and this temperature is well tolerated for several hours [15]. Heating allows the elimination of variations in local circulation induced by patient anxiety, pain, and PaCO_2_, which largely control vascular tone. 

A normal TcPO2 is considered around 60 mmHg whatever the electrode location [16]. Interestingly, TcPO2 can be used whatever the etiology of extremities’ arterial insufficiency. Indeed, no significant difference was found in the TcPO2 indices between arteriosclerosis and thromboangiitis obliterans, nor among different sites of peripheral vascular occlusion [8]. Also, the performances of TcPO2 measurement were found to be similar in diabetic and non-diabetic patients [17]. Similarly, body mass index variation did not affect TcPO2 measurements [18].

Overall, the TcPO2 values are affected by numerous factors including temperature in the tissues, the degree of oxygen metabolism in tissues, the circulatory status, the peripheral blood perfusion, and the local skin and anatomical conditions [19]. Measurement at the preferred amputation level is preferred but not always possible when the epidermis is thin (because of edema and inflammation). Bony prominences, such as superficial vein and tendon, should be avoided as they restrict the capillary blood flow. Furthermore, it is widely accepted that measurements in a sitting position with the lower extremity in a vertical position can lead to false positive results. Conversely, smoking and coffee consumption, pain, and anxiety may lead to vasoconstriction and under-estimate TcPO2 value (Table 1).

Finally, evaluators should keep in mind the reproducibility. The reliability of TcPO2 measurements was evaluated in ten elderly normal subjects: the TcPO2 values were obtained on three separate occasions at two-week intervals, and the measurement-to-measurement variation averaged at 8 mmHg. The size of the confidence intervals could be reduced substantially by taking the mean of two or more TcPO2 measurements taken at separate times [20].

Similarly, an older study published in 1978 by Gothgen and Jacobsen focused on 20 patients between 19 and 80 years scheduled for Ear-Nose-Throat or eye surgery and without pulmonary and cardiovascular symptoms or signs. Two electrodes, heated to 43 °C, were placed symmetrically at the level of the third rib on the right and left mid-clavicular lines. TcPO2 measurements were recorded every 5 min after stabilization and electrodes were switched. Results did not differ significantly between sides and with time. The standard deviation between measurements at the same location was between 0 and 15 mmHg, and the 95% confidence interval did not exceed ±7.5 mmHg [21]. This standard deviation corresponds to the accuracy found for oxygen tension measurements in automatic blood gas analyzers and was at this time accepted for clinical use [22].

## 4. Consequences of Major Amputation Levels on Patients Mobility

Besides primary wound healing without revision surgery, the main concern is to regain mobility. Different authors have used various ways to define successful mobility, from being able to wear a prosthesis, to transfer (bed to chair) themselves (which gives them some autonomy), to walk at home, and to walk outside [23,24,25,26,27]. It is therefore not easy to compare success between studies and through the years, but from 1986 to 2016, most studies including patients with arterial disease reported a mobility success around 30–40% [23,24,25,26,27].

In case of major amputation, it is commonly admitted that the best prognosis of rehabilitation is encountered in case of preservation of the knee joint. Older literature^25^ did not reveal whether AKA or BKA have functional outcomes, but in recent years, there is increasing literature reporting functional superiority of BKA. 

In a population of 4965 nursing home residents, BKA amputation was found to have a superior functional trajectory compared with AKA [28].

Furthermore, Taylor et al. investigated the relationship between preoperative clinical characteristics and post-operative functional outcomes. A total of 627 lower-limb major amputations (37.6% BKA, 4.3% through knee amputations, 34.5% AKA, and 23.6% bilateral amputations) were performed on 533 patients, 91.5% of whom suffered from vascular disease. One year after amputation, 55% of patients were able to maintain ambulation. AKA was a statistically significant preoperative factor independently associated with not wearing a prosthesis. Others significant factors independently associated with not being able to wear a prosthesis (and therefore not being able to regain mobility) were being non-ambulant or homebound before amputation, being older than 60 years, the presence of dementia, end stage renal disease, and coronary artery disease [29]. Therefore, many patients will never be able to walk on an artificial limb and a realistic appraisal may judge in these cases a wheelchair to be more appropriate than a prosthesis. This knowledge might influence the selection of the amputation level.

## 5. Selection of the Amputation Level

A variety of criterions need to be considered in the choice of an amputation level [30]. Besides the amount of necrotic or infected tissues, perfusion majorly influences the amputation level selection. In the past, clinicians have used features of the physical examination including color, temperature, the most distal pulse, and wound edge bleeding to assess the potential for healing at a particular level. These clinical criteria are clearly helpful but have some limitations in the prediction of healing.

### 5.1. General Predictors of Healing Success

Because of the generalized nature of atherosclerosis, patients with limb ischemia also suffer from hypertension, cardiac ischemia, and cerebrovascular disease. Optimizing treatment for these conditions is desirable and may actually improve the peripheral circulation. Patients undergoing amputations are also nutritionally disadvantaged. It has been shown that serum albumin levels are associated with success or failure of the healing procedure [31]. Preoperative treatment of infected lesions in the relevant limb has also been shown to reduce the subsequent morbidity [32].

### 5.2. TcPO2 Threshold Values for the Determination of the Amputation Level

Choosing the optimal level for amputation is therefore a clinical, serious issue affecting both the functional and vital prognosis of the patient. 

In 1983, Dowd et al. found a better correlation in 24 lower-limb amputee patients between amputation healing and the determination of the amputation level determined based on TcPO2 measurements rather than based on clinical exam [33]. In 1987, Malone et al. showed, for 48 patients undergoing 52 elective major lower extremity amputations, that TcPO2 value and foot-to-chest TcPO2 index were significantly reliable for predicting healing of trans-metatarsal, above, and below-knee amputations, while ankle-brachial index and absolute popliteal artery Doppler systolic pressure values were not [34]. 

Early studies defined amputation success as stump healing. Several studies indicated that calf TcPO2 values greater than 40 mmHg were associated with a high percentage of successful wound healing after BKA [15,33,35,36,37], whereas values lower than 20 mmHg indicated an increased risk of unsuccessful healing [10,34,38,39]. However, these threshold values have limited clinical applicability because (1) most patients with significant LE-PAD have calf TcPO2 values lower than 40 mmHg [40,41], (2) successful wound healing after BKA occurs in a significant percentage of patients despite calf TcPO2 values lower than 20 mmHg [11,36,39], and (3) TcPO2 measurements are dependent on the systemic blood flow and arterial oxygen content, which are dependent on the patient’s cardiac output, arterial pO_2_, hemoglobin level, temperature, and P_50_ [42].

In 1995, Wütschert and Bounameaux analyzed the literature data by means of receiver operating characteristic curve in order to determine a critical TcPO2 level for which the rate of stump failure would be clinically acceptable (defined as 20% of stump failure). A total of 10 prospective studies with blind evaluation of TcPO2 were included, representing 615 patients. No threshold value below which the stump definitely failed to heal could be identified. Authors recommend 20 mmHg as the optimal cutoff, which would predict healing with 80% accuracy. They notice that the cutoff should never be set above 30 mmHg because of the rapid decrease of the negative value and accuracy beyond this threshold, in the absence of any increase of the positive predictive power [43]. 

More recently, an amputation success has been defined as an amputation that does not require amputation revision. Until now, only one study performed a multiple logistic regression analysis to analyze independent risk factors associated with re-amputations. TcPO2 captors were placed on 22 patients at the level of forefoot, ankle, below the knee, through the knee, and above the knee, on both lateral and medial sides. Amputations were performed at the level where one of the two measurements (lateral or medial) was >30 mmHg or where both were >20 mmHg. These results were compared to a historical cohort of 118 patients without TcPO2 measurements. The number of re-amputations was significantly higher in the case group than in the historical cohort (about twice as much). The selection of the amputation level at a TcPO2 of 30 mmHg resulted in a positive predictive value (PPV) of re-amputation of 41% and a negative predictive value (NPV) of 90%. A cut-off value of 20 mmHg resulted in a PPV of 41% and a NPV of 77%, authors observed [44]. 

A meta-analysis was conducted in 2012 to determinate the validity of TcPO2 use as a predictor of lower limb amputation healing. A total of 31 studies met the inclusion criteria. However, due to heterogeneity, only 14 prospective cohort studies were compatible with the calculation of an unadjusted relative risk of occurrence of amputation revision. These 14 studies included 626 patients with 658 amputations. The relative risks of re-amputation were similar for different TcPO2 cut-off values (10 mmHg: OR = 1.80, 95% CI [1.19; 2.72]; 20 mmHg: OR = 1.75, 95% CI [1.27; 2.40]; 30 mmHg: OR = 1.41, 95% CI [1.22; 1.62]; 40 mmHg: OR = 1.24, 95% CI [1.13; 1.39]). Authors concluded evidence were insufficient to judge whether TcPO2 adds relevant information beyond clinical data and to suggest an optimal TcPO2 threshold value [45].

In 2016, a review performed by Nishio et al. included 11 studies describing the association between TcPO2 and wound healing after amputation of ischemic limb. The analysis demonstrated that the optimal TcPO2 cut-off value enabling the healing of amputation stumps while avoiding unnecessary large amputation was approximately 20 mmHg. However, because definitive post-amputation healing is desirable, they considered a higher cut-off value of 30 mmHg [46].

Some authors have suggested that, because of tremendous progress made in endovascular therapy, current patients are different from the ones included in historical studies, challenging the relevance of their findings. More recently, Columbo et al. found TcPO2 levels ≥ 40 mmHg as significantly associated with successful one-year amputation healing in a study including 120 patients and 130 BKA. The one-year freedom of conversion to AKA was decreased compared to those with lower TcPO2 values (60% TcPO2 < 40 mmHg vs. 81% TcPO2 ≥ 40 mmHg; *p* = 0.04) [27]. 

In another study, authors reviewed 303 amputations in 11 years (42 amputations in the mid-foot, 7 ankle, 154 leg, 28 knee joint, and 75 AKA), authors found that TcPO2 of 35 mmHg did not discriminate between success and stump failure. TcPO2 was measured on the dorsum of the foot and 10 cm below the knee. In multivariate analysis adjusting for case-mix, TcPO2 values were unrelated to stump failure, except for thigh amputations [47].

Berli et al. also reviewed 180 patients undergoing major lower extremity amputation including 154 BKA. The probe was applied at the planned site of amputation, 12 to 15 cm below the knee joint line for below-knee amputations, while the patient was lying flat. They concluded that TcPO2 < 40 mmHg is useful for predicting wound healing problems [48].

Therefore, there is to date no consensus regarding TcPO2 value to precisely define a level of amputation. 

## 6. Current Recommendations?

Until now, no society has endorsed formal guidelines regarding the threshold of TcPO2 that should be used for choosing the amputation level. François Becker proposed a critical threshold between 20 and 30 mmHg in a chapter [49] endorsed by the Société Francaise de Médecine Vasculaire (French Society of Vascular Medicine), the Collège des Enseignants de Médecine Vasculaire (College of Vascular Medicine), and the Collège Francais de Pathologie Vasculaire (French College of Vascular Diseases).


There is no precise recommendation of the electrode placement. Thereafter, should the electrode be placed on the medial, lateral, or two faces of the limb? How far from the joints should the electrode be placed? All studies previously cited and described different protocols, but each center should be encouraged to follow a standardized protocol and to, at the very minimum, describe carefully electrode placement site in reports.


## 7. How Can TcPO2 Performance Be Improved?

### 7.1. Control Site

Many authors proposed to consider measurement on a control site to take into account potential central circulatory failure. The control site is, consensually, 5 cm under the medio-clavicular line. Mustapha et al. found no greater advantage of calculating a ratio with a control site than using an absolute value [13]. Nevertheless, TcPO2 values of the chest are known to decrease with advancing age [8].

Kram et al. performed TcPO2 measurements on the anterolateral calf and posteromedial calf in 40 patients before BKA, while choosing the middle of the upper arm as control site. Calculation of a critical Po2 index (defined as the lesser of the anterior or posterior calf/brachial TcPO2 ratios) resulted in improved predictive accuracy; indeed, all six patients with a critical Po2 index of 0.20 or less had unsuccessful wound healing after BKA, whereas all but one of the 34 patients with a critical Po2 index greater than 0.20 had successful healing. The use of a critical Po2 index greater than 0.20 as predictive factor of successful healing after BKA was associated with a sensitivity of 100%, a specificity of 86%, and an overall accuracy of 98%, whereas these values were respectively 82%, 86%, and 83% when absolute calf TcPO2 values were used [50]. The authors chose brachial rather than chest TcPO2 measurement because of its greater ease of standardization (midway between the axilla and the antecubital fossae on the anterior skin, overlying the biceps muscle).

### 7.2. 100% Oxygen Inhalation

After 10 min of 100 % oxygen inhalation, Mustapha el al. observed a significant increase in TcPO2 values at the control site and at the 10 cm below-knee site in 20 patients with 23 ischemic lower limbs [51]. These results were interesting for the borderline TcPO2 readings, and were increasing the chances of a successful BKA. In a prospective double-blind study including 101 patients undergoing 119 limbs amputations, TcpO2 measurements were obtained from the dorsum of the foot and 10 cm distal to the patella, both prior to and 10 min after inhalation of 100% oxygen. An initial TcPO2 value greater than 10 mmHg or an increase greater than 10 mmHg after oxygen inhalation were considered as predictor of a successful outcome. In the BKA group (57 patients), the test was 95% sensitive, 100% specific, and 95% accurate [39].

### 7.3. Elevation Measurements

Bacharach et al. assessed TcPO2 before amputations of 90 lower limbs in 75 patients. Electrodes were placed along the limb (1) on the dorsum of the foot over the metatarsals, (2) 4 cm above the medial malleolus, (3) 8 cm below the knee, and (4) 10 cm above the knee. Measurements were initially made while the patient was in supine position and then repeated after elevation of the lower extremities to 30° for 3 min. Electrode placement sites were used to obtain TcPO2 values for the corresponding site: thigh for above the knee amputations, calf for below the knee amputation, and foot for trans-metatarsal and Syme’s amputations. A TcPO2 ≥ 40 mmHg was associated with primary or delayed healing in 51 of 52 limbs (98%). Delayed healing occurred beyond six weeks with or without local debridement procedures. A TcPO2 value < 20 mmHg was always associated with failure. For patients with a TcPO2 between 20 and 40 mmHg, limb elevation measurements improved the predictability of outcome (mean decrease of 4.0 mmHg in the healed, 9.4 mmHg in the delayed, and 14.8 mmHg in the failed groups) [52].

Elevation measurements are not feasible in patients who are in pain when at rest, which is usually the case in patients for whom an amputation is required.

## 8. Conclusions

TcPO2 is a precise reflection of the skin perfusion in optimal conditions of measurements. Surgeons must take into account all confounding factors to interpret TcPO2 values, mostly edema. Importantly, TcPO2 values should not be used as the sole criterion to define lower limb amputation level [43], but always be evaluated together with the patient’s characteristics (angiography), clinical judgement, and comorbidities (coronary artery disease, cerebrovascular disease, mobility before amputation) [53].

No consensus on TcPO2 threshold that could guarantee amputation healing is currently available. However, one should keep in mind that strategy behind the use of TcPO2 is to offer the most distal amputation level in spite of pejorative clinical criteria of ischemia.

Data are missing to evaluate whether clinicians should use absolute or relative values of TcPO2, as well as (in case of use of relative values) the position of the control site, the place of the oxygen inhalation test, and elevation measurements in current practice, irrespective of the time spent for the measurements to be performed.

## Figures and Tables

**Table 1 jcm-10-01413-t001:** Conditions affecting transcutaneous oximetry (TcPO2) reliability.

	Under Estimation	Over Estimation
Skin quality	Cutaneous sclerosis	Inflammation
	Edema	
Room temperature	<22 °C	>24 °C
	Air flow (door open)	
Electrode location	Bone prominence	
	Superficial tendon	
	Superficial vein	
Patient-related	Patient talking	Sitting (over decubitus)
	Pain	
	Anxiety Smoking or coffee < 2 h

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
