# Peer review of "What Is Currently the Role of TcPO2 in the Choice of the Amputation Level of Lower Limbs? A Comprehensive Review"

_jcm, 2021, doi:10.3390/jcm10071413_

Round 1

Reviewer 1 Report

Dear authors, thank you for the improvements made in this revised manuscript. There are still some minor grammar and style mistakes which can easily be corrected. However, there is one major point of concern that needs to be corrected. I will start with this major point:

  • Lines 46-47 // 155-157:  The statement that data are missing on mobility success between AKA/BKA is plain and simple false. The literature that you have cited is from 1984 and 1992. There are recent papers that clearly state superiority of BKA compared to AKA in terms of functional outcome. You need to change both statements in the abstract and in the main text. It is possible to write something like "Literature of the 1980` s and 1990 `s did not reveal whether AKA or BKA have better functional outcomes but in recent years, there is  increasing literature reporting functional superiority of BKA". In addition to prior literature I found (exemplary results of literature only, not concluding):
    1. Shane R. Wurdeman, Phillip M. Stevens & James H. Campbell (2020) Mobility analysis of amputees (MAAT 3): Matching individuals based on comorbid health reveals improved function for above-knee prosthesis users with microprocessor knee technology, Assistive Technology, 32:5, 236-242, DOI: 10.1080/10400435.2018.1530701
    2. Batten HR, McPhail SM, Mandrusiak AM, Varghese PN, Kuys SS. Gait speed as an indicator of prosthetic walking potential following lower limb amputation. Prosthetics and Orthotics International. 2019;43(2):196-203. doi:10.1177/0309364618792723
    3. Sansam K, Neumann V, O'Connor R, Bhakta B. Predicting walking ability following lower limb amputation: a systematic review of the literature. J Rehabil Med. 2009 Jul;41(8):593-603. doi: 10.2340/16501977-0393. PMID: 19565152.

Minor changes:

  • Line 121-123: The sentence is unclear. Rephrase: "Measurement at the preferred amputation level is preferred but not always possible, when the epidermis ist thin (because of edema and inflammation)."
  • Lines 259-261: I think there is no affiliation of one of the authors to Zurich University Hospital. It is from Geneva University Hospital. I would leave any affiliation out of the main text. Simply cite the first authors name or use "another study reported".
  • Line 278: Style and Grammar: Rephrase "There is no precise recommendation of..."
  • Line 332: Style and Grammar: Rephrase "Surgeons must take into account all confounding factors..." 
  • Line 335: Style and Grammar: Rephrase "....clinical judgement, and comorbidities (....)."
  • Lines 338-339: Rephrase: "However, one should keep in mind that the strategy behind the use of TcPO2 is to offer the most distal amputation level in spite of...."

Author Response

 Dear Reviewer 1,

We warmly thanks reviewer 1 to accept to review another time our work very carefully. We are now convince that mobility is better after BKA than after AKA and we delate the paragraph about the old  study of Oxford surgeons and also correct introduction and text as proposed.

Minor Revisions were also changed in the text (in yellow).

If reviewer 2 agree, we will ask to editor if it is possible to add Reviewer to authors.

Reviewer 2 Report

The TcPo2 values does not provide substantial informations in patients with asymptomatic obstructions (Fontaine stage I) or intermittent claudication (Fontaine stage II) but is of clinical impact in limbs with rest pain (stage III) and skin lesions (stage IV). In macrocirculatory patients TcPo2 values depend on PAOD staging, haemodynamic compensation and calf artery patiens. 

The authors provide a review about the role of TcPO2 in the choice of the amputation level of the lower limb. I offer a few thoughts and suggestions:

  1. Please clarify the relationship between site of vascular obstruction and TcPO2 values. 
  2. A focus on smokers and elevated BMI is encouraged
  3.  Please specify how TcPO2 is linked to mortality

Author Response

Dear Rewiewer,

thank you for your interest in our work. 

Like reviewer 2, we think that TcPo2 values is of clinical impact in limbs with rest pain (stage III) and skin lesions (stage IV) which unfortunately lead sometimes to amputation.

1.The site of vascular obstruction is a complementary data used by surgeons with TcPO2 to select amputation level but is not always related toTcPO2 value depending on hemodynamic compensation and collateral circulation development.

We add a sentence: “TcPO2 value depends on LE-PAD Fontaine classification, site of vascular obstruction and haemodynamic compensation by calf arteries patency.”

2.We found only one study about BMI and TcPO2. We add this sentence: “Similarly, body mass index variation did not affect TcPO2 measurements.”

We also precise risk of under estimation of TcPO2 value with tobacco, coffee, pain or anxiety as describe in Table 1. “Conversely, smoking and coffee consumption, pain and anxiety may lead to vasoconstriction and to underestimate TcPO2 value”.

3.Amputation is associated with substantially high mortality. Considering that TcPO2 value is linked to PAOD staging and to amputation outcomes, poor TcPO2 value is also linked to mortality. Mortality is almost due to comorbidities. We also think that this is a major point and we stated this fact in the introduction

This manuscript is a resubmission of an earlier submission. The following is a list of the peer review reports and author responses from that submission.

Round 1

Reviewer 1 Report

  1. General comments:
    (a) Comprehensive structure.
    (b) English language needs to be improved
    (c) References are not up to date
  2. Motility: most authors use "mobility" rather than "motility". I would suggest to replace motility throughout the manuscript.
  3. Specific remarks:

  • Lines 39-41: This statement is at least debatable. In the surgical literature, a major amputation is one above the ankle joint. Minor amputations are defined as amputations up until the ankle level (=Syme level). A reference is missing.
  • Lines 44-46: This is a bold statement. There is evidence that BKA outperform AKA in terms of energy expenditure and walking ability.
  • Lines 91-93: Please comment on the costs of the TCPO2 measurement device. In my environment a new device costs on average 27730 Euros. Admittedly, this is rather easily feasible for an European country but what about poorer environments?
  • Lines 91-93: replace "capital outlay" by "costs"
  • Line 99: replace "operator" by "surgeon".
  • Lines 98-100: This is a misleading statement. The surgeon usually places the electrode at usual amputation levels from distal to proximal considering the amount of infected tissue or necrotic tissue. Please clarify.
  • Lines 111-116:
    (a) It is widely accepted that measurements in a sitting position or with the lower extremity in a vertical position can lead to false positive results. Table 1 contains this information but it is too important to not mention it in the main text. Please incorporate body positioning into the main text.
    (b) It is rather beneficial to measure where the surgical wound needs to heel then where it is easy to perform measurements. Rephrase the sentence 113-115 accordingly.
  • Line 133: replace "motility" with "mobility" and throughout the rest of the manuscript
  • Misleading title: rather "Consequences of major amputation levels on patient mobility"
  • Lines 134-138: This is an absolutely confusing paragraph:
    (a) line 134: Rephrase, the sentence does not make sense at all. Did you mean: Besides primary wound healing without revision surgery, the main concern is to regain mobility? Rephrase.
    (b) 135-137: References are missing
    (c) Lines 137-138: This would be devastating if ALL major amputation would have such bad outcomes. Clarify the patient collective! You can use TCPO2 in young trauma patients, too... Those don't have mobility success of only 40% after major amputation. Then, we are in 2021! Why do you cite literature until 2016 only?
    (d) Line 139: Rephrase the sentence and start with "In case of major amputations...." Otherwise it is unclear that you refer to major amputations only.
  • Lines 139-14: another bold statement. Read the following literature and add it to this particular paragraph, then mitigate this particular sentence.

    Vogel TR, Petroski GF, Kruse RL. Impact of amputation level and comorbidities on functional status of nursing home residents after lower extremity amputation. Journal of Vascular Surgery 2014;59(5):1323-1330.e1. doi: https://doi.org/10.1016/j.jvs.2013.11.076.

    Esquenazi A. Gait analysis in lower-limb amputation and prosthetic rehabilitation. Phys Med Rehabil Clin N Am2014;25(1):153-67. doi: 10.1016/j.pmr.2013.09.006.
  • Lines 161-166: There is so much more that needs to be considered when the amputation level is being selected (prior mobility of the patient, prior ipsi- or contralateral amputation, comorbidities, age, patients wish) that I suggest to you to rephrase and simplify this paragraph:

    Suggestion:
    "A variety of criterions need to be considered in the choice of an amputation level (add your reference). Besides the amount of necrotic or infected tissues, perfusion majorly influences the amputation level selection." Continue with "In the past..."
  • Lines 177-236: you seem to have stopped your literature research in 2016. Read the following literature and add it to this paragraph:

    Zingg M, Lacraz A, Robert-Ebadi H, Kressmann B, Glauser F, Waibel F, et al. Transcutaneous Oxygen Pressure Values Often Fail to Predict Stump Failures after Foot or Limb Amputation in Chronically Ischemic Patients. Clin Surg. 2019; 4: 2366.

    Berli MC, Wanivenhaus F, Kabelitz M, Götschi T, Böni T, Rancic Z, Waibel FWA. Predictors for reoperation after lower limb amputation in patients with peripheral arterial disease. Vasa. 2019 Aug;48(5):419-424. doi: 10.1024/0301-1526/a000796. Epub 2019 May 7. PMID: 31063045.

  • Line 293: Replace "prescribers" with "surgeons"
  • Line 293: You cannot start to list "edema" as a major confounder (you even state "mostly") in the conclusions. Add edema in the corresponding paragraph as a major confounder of measurement accuracy.
  • Line 296: spelling mistake: "judgement"
  • Line 296: change to: ...judgement, and comorbidities (....)

Author Response

Dear Reviewer,

We warmely thank reviewer for his carefully reading.

  • Lines 39-41: This statement is at least debatable. In the surgical literature, a major amputation is one above the ankle joint. Minor amputations are defined as amputations up until the ankle level (=Syme level). A reference is missing.

We corrected the definitions according the surgical literature and add a reference.

  • Lines 44-46: This is a bold statement. There is evidence that BKA outperform AKA in terms of energy expenditure and walking ability.
  • Lines 91-93: Please comment on the costs of the TCPO2 measurement device. In my environment a new device costs on average 27730 Euros. Admittedly, this is rather easily feasible for an European country but what about poorer environments?

Indeed, we agree with reviewer that TcPO2 is not cost effective in poor environnements, especially as healing need to be happen faster to avoid others complications. We add in the text that TcPO2 is well accessible in developed countries.

  • Lines 91-93: replace "capital outlay" by "costs"

We replace word as reviewer propose.

  • Line 99: replace "operator" by "surgeon".

We disagree on this point as, in our hospital, electrode place is choosen by vascular doctor or by nurses practioners. Operator is common term to design habits all over the world.

  • Lines 98-100: This is a misleading statement. The surgeon usually places the electrode at usual amputation levels from distal to proximal considering the amount of infected tissue or necrotic tissue. Please clarify.

We clearly agree with reviewer but this is the general principle. In fact, it does not define precise positions. For example, where is the better position on the limb ? on medial face ? lateral face ? posterior face ? how distant from the knee ? one electrode at every stage or two ? We tried to clarified this statement in the text adding « exact » position.

  • Lines 111-116:
    (a) It is widely accepted that measurements in a sitting position or with the lower extremity in a vertical position can lead to false positive results. Table 1 contains this information but it is too important to not mention it in the main text. Please incorporate body positioning into the main text.

We agree with review and add this sentence in the text.
(b) It is rather beneficial to measure where the surgical wound needs to heel then where it is easy to perform measurements. Rephrase the sentence 113-115 accordingly.

We review with reviewer and add in the text : « Measurements where a wound needs to heal does not always allow to measure on skin where the epidermidis is thin (because of edema and inflammation). »

  • Line 133: replace "motility" with "mobility" and throughout the rest of the manuscript

We replace motiliy with mobility in the text.

  • Misleading title: rather "Consequences of major amputation levels on patient mobility"

We corrected the title according.

  • Lines 134-138: This is an absolutely confusing paragraph:
    (a) line 134: Rephrase, the sentence does not make sense at all. Did you mean: Besides primary wound healing without revision surgery, the main concern is to regain mobility? Rephrase.

We corrected the sentence.
(b) 135-137: References are missing
(c) Lines 137-138: This would be devastating if ALL major amputation would have such bad outcomes. Clarify the patient collective! You can use TCPO2 in young trauma patients, too... Those don't have mobility success of only 40% after major amputation. Then, we are in 2021! Why do you cite literature until 2016 only?

We precised that these results mostly concerned patients with arterial disease.
(d) Line 139: Rephrase the sentence and start with "In case of major amputations...." Otherwise it is unclear that you refer to major amputations only.

We added « in case of major amputations » at the  begining of the sentence.

  • Lines 139-14: another bold statement. Read the following literature and add it to this particular paragraph, then mitigate this particular sentence.

    Vogel TR, Petroski GF, Kruse RL. Impact of amputation level and comorbidities on functional status of nursing home residents after lower extremity amputation. Journal of Vascular Surgery 2014;59(5):1323-1330.e1. doi: https://doi.org/10.1016/j.jvs.2013.11.076.

    Esquenazi A. Gait analysis in lower-limb amputation and prosthetic rehabilitation. Phys Med Rehabil Clin N Am2014;25(1):153-67. doi: 10.1016/j.pmr.2013.09.006.

We thanks reviewer for this literature that we missed. We corrected sentence with replace « data are missing » with « data are divergent ». We added Vogel et al. results after those of oxford surgeons. Vogel results are not easy to report because they compare functionnal status and not mobility.

  • Lines 161-166: There is so much more that needs to be considered when the amputation level is being selected (prior mobility of the patient, prior ipsi- or contralateral amputation, comorbidities, age, patients wish) that I suggest to you to rephrase and simplify this paragraph:

    Suggestion:
    "A variety of criterions need to be considered in the choice of an amputation level (add your reference). Besides the amount of necrotic or infected tissues, perfusion majorly influences the amputation level selection." Continue with "In the past..."

We thanks reviewer this proposition and replace the sentences according.

  • Lines 177-236: you seem to have stopped your literature research in 2016. Read the following literature and add it to this paragraph:

Zingg M, Lacraz A, Robert-Ebadi H, Kressmann B, Glauser F, Waibel F, et al. Transcutaneous Oxygen Pressure Values Often Fail to Predict Stump Failures after Foot or Limb Amputation in Chronically Ischemic Patients. Clin Surg. 2019; 4: 2366.

Berli MC, Wanivenhaus F, Kabelitz M, Götschi T, Böni T, Rancic Z, Waibel FWA. Predictors for reoperation after lower limb amputation in patients with peripheral arterial disease. Vasa. 2019 Aug;48(5):419-424. doi: 10.1024/0301-1526/a000796. Epub 2019 May 7. PMID: 31063045.

Recent litterature published in 2019 concerned retrospective data and do not modify the point of view about TcPO2 in the selection of amputation level. However, we added this literature to be comprehensive.

  • Line 293: Replace "prescribers" with "surgeons"

We replace prescribers with surgeons

  • Line 293: You cannot start to list "edema" as a major confounder (you even state "mostly") in the conclusions. Add edema in the corresponding paragraph as a major confounder of measurement accuracy.

We added edema in the corresponding paragraph.

  • Line 296: spelling mistake: "judgement"

We corrected the mistake.

  • Line 296: change to: ...judgement, and comorbidities (....)

We corrected the mistake.

Reviewer 2 Report

The paper is well researched, impressively so, and exhaustive. My main criticism is the title and purpose state the purpose of reviewing the use of TcpO2 in amputation site selection, but the paper is interrupted by several paragraphs on AKA vs. BKA, sleep apnea, and oxygen challenge.  These are interesting topics, but unfortunately distract from the intended message.

Several specific notes. 

Page 2 Line 89:  the words "admitted early on" should be replaced by "shown"

Page 5 Lnae 205:  "resumption" should be "revision"

Page 7 Line 282 to 289:  the outcomes have been misquoted.  It appears that the sentences were edited and a phrase was lost in the process. I suggest the authors revise.

Page 7 Line 298:  "consensual" should be "consensus on"

Over all English is good, with a few odd phrases that could easily be corrected. However the reading level of the writing is lower than expected for a medical journal

I suggest focusing the paper with the above edits.  A statement on how the procedure is used by the authors would be appropriate.

Author Response

Dear Reviewer,

We warmely thank reviewer for his carefully reading.

The paper is well researched, impressively so, and exhaustive. My main criticism is the title and purpose state the purpose of reviewing the use of TcpO2 in amputation site selection, but the paper is interrupted by several paragraphs on AKA vs. BKA, sleep apnea, and oxygen challenge.  These are interesting topics, but unfortunately distract from the intended message.

 We understand that reader could be distracted by the paragraph about AKA vs BKA but it appears to be very important to consider the mobility rate according the level of amputation choosen by the surgeon which is, according us, a major outcome after healing. Furthermore, one of our objectives is to review how to improve TcPO2 performances and oxygen challenge is one of this ways.

However, paragraph on sleep apnea is not essential and has been removed.

Several specific notes. 

Page 2 Line 89:  the words "admitted early on" should be replaced by "shown"

We replace this words.

Page 5 Lnae 205:  "resumption" should be "revision"

We replace this word according.

Page 7 Line 298:  "consensual" should be "consensus on"

We replace this word accordingly.